# Long Non-Coding RNA *H19* Prevents Lens Fibrosis through Maintaining Lens Epithelial Cell Phenotypes

**DOI:** 10.3390/cells11162559

**Published:** 2022-08-17

**Authors:** Lang Xiong, Yan Sun, Jingqi Huang, Pengjuan Ma, Xiaoran Wang, Jiani Wang, Baoxin Chen, Jieping Chen, Mi Huang, Shan Huang, Yizhi Liu

**Affiliations:** State Key Laboratory of Ophthalmology, Zhongshan Ophthalmic Center, Sun Yat-sen University, Guangdong Provincial Key Laboratory of Ophthalmology and Visual Science, Guangzhou 510060, China

**Keywords:** long non-coding RNA *H19*, lens fibrosis, epithelial-mesenchymal transition, TGF-β2, Smad-dependent signaling

## Abstract

The integrity of lens epithelial cells (LECs) lays the foundation for lens function and transparency. By contrast, epithelial-mesenchymal transition (EMT) of LECs leads to lens fibrosis, such as anterior subcapsular cataracts (ASC) and fibrotic forms of posterior capsule opacification (PCO). However, the underlying mechanisms remain unclear. Here, we aimed to explore the role of long non-coding RNA (lncRNA) *H19* in regulating TGF-β2-induced EMT during lens fibrosis, revealing a novel lncRNA-based regulatory mechanism. In this work, we identified that lncRNA H19 was highly expressed in LECs, but downregulated by exposure to TGF-β2. In both human lens epithelial explants and SRA01/04 cells, knockdown of *H19* aggravated TGF-β2-induced EMT, while overexpressing *H19* partially reversed EMT and restored lens epithelial phenotypes. Semi-in vivo whole lens culture and *H19* knockout mice demonstrated the indispensable role of *H19* in sustaining lens clarity through maintaining LEC features. Bioinformatic analyses further implied a potential *H19*-centered regulatory mechanism via Smad-dependent pathways, confirmed by in vitro experiments. In conclusion, we uncovered a novel role of *H19* in inhibiting TGF-β2-induced EMT of the lens by suppressing Smad-dependent signaling, providing potential therapeutic targets for treating lens fibrosis.

## 1. Introduction

The ocular lens is a unique transparent organ whose homeostasis is maintained mainly by a single layer of lens epithelial cells (LECs) underneath the anterior lens capsules [1]. Without proper functioning of LECs, lens transparency cannot be sustained; instead, opacification of the lens canH lead to cataracts, which are the leading cause of vision loss across the globe [2].

Upon exposure to external insults, such as ocular trauma or surgeries, transforming growth factor-beta (TGF-β), which is usually latent [3] and tightly regulated in the lens, increases dramatically to a great extent [4,5]. In particular, TGF-β2 predominates in human ocular media [6,7,8,9]. The abnormally activated TGF-β signaling leads to the pathological transformation of LECs to mesenchymal cells with matrix deposition and capsule wrinkling, contributing to epithelial-mesenchymal transition (EMT). During this process, LECs lose their apicobasal cell polarity and intercellular junctions, acquire mesenchymal characteristics, and transdifferentiate into multi-layered spindle-shaped myofibroblasts with an accumulation of α-smooth muscle actin (αSMA) [10,11]. These cells are prone to proliferate and migrate aberrantly, accompanied by an abnormal secretion of fibronectin (FN) and collagen I/III, not typically expressed in the lens, into the extracellular matrix (ECM), driving the contraction of the lens capsules [12,13]. Importantly, EMT lays the foundation for fibrotic cataracts, including fibrotic forms of posterior capsular opacification (PCO) and anterior subcapsular cataracts (ASC) [4,5,8,9,14].

Accumulative studies have pointed out that TGF-β2-induced EMT is a crucial feature manifested in embryonic development and various pathological phenomena, particularly wound healing, fibrotic diseases, and some invasive cancers [15,16,17]. EMT triggers lens fibrosis and can be targeted at multiple levels [18,19,20]. Long non-coding RNAs (LncRNAs) are a group of RNAs whose length exceeds 200 nucleotides but cannot be translated into functional proteins [21]. Instead of serving as genetic background noises, they emerge as a breakthrough for deciphering the molecular mechanism of both biological and pathological processes, including EMT [22,23,24]. More importantly, their mechanisms of action are diverse and still up-and-coming: lncRNAs can epigenetically regulate gene expression through binding to DNA or chromatin modifiers [25,26]. They can also sequestrate microRNAs (miRNAs) or act as a scaffold, decoy, or guide for vital proteins [27]. Notably, *H19* is the first discovered eukaryote lncRNAs with a relatively high evolutionary conservancy in humans and mice [28,29], and plays an essential role in human embryonic development and diverse conditions, including invasive cancers and fibrotic diseases [30,31]. However, its role in the pathogenesis of fibrotic cataracts remains unclear.

Using cutting-edge RNA-sequencing, this study first identified that *H19* was highly expressed in LECs but downregulated upon TGF-β2 treatment. Then, both loss- and gain-of-function studies conducted in human lens epithelial explants and cell lines revealed that *H19* could help maintain LECs properties and reverse TGF-β2-induced EMT in the lens. In addition, semi-in vivo whole lens culture studies showed that when *H19* was forcibly knocked down, lens transparency could not be sustained following the loss of LEC features. Lens opacities similar to ASC were also observed in homozygous *H19* knockout mice. Furthermore, we explored possible downstream mechanisms responsible for its functions, highlighting the potential role of the *H19/miR-675-3p/SMAD4* axis in fibrotic cataracts. To conclude, this study revealed *H19* as an indispensable modulator of TGF-β2-induced EMT in the lens, implying that *H19* could be a novel prophylactic or therapeutic target for tackling human fibrotic cataracts.

## 2. Materials and Methods

### 2.1. Human Lens Epithelial Explants Collection, Cell Culture, and Treatment

All human samples collected from organ donors were provided by the Guangdong Provincial Eye Bank. We first obtained written informed consent from the family members of organ donors, for each human subject involved. This study was approved by the institutional review board of the Zhongshan Ophthalmic Center (Project identification code: 2019KYPJ118; Date of approval: 20 September 2019) and conducted following the principles of the Declaration of Helsinki. The ages of donors ranged from 30 to 50 years old. With capsulotomy scissors dissecting the lenses at the equatorial regions and all lens fibers removed, the remaining anterior lens capsules with underlying lens epithelial cells were then collected and annotated as human LEEs. Immediately after collection, they were cultured in Minimum Essential Medium (MEM, Thermo Fisher Scientific, Waltham, MA, USA) containing 1% fetal bovine serum (FBS, Thermo Fisher Scientific, Thornton, AU) with 1%NEAA (Life Technologies, Carlsbad, CA, USA), as previously described [32]. For the TGF-β2-induced EMT model, the human lens epithelial explants were first cultured for at least 6 h to ensure the excellent condition of the LECs, and then treated with 10 ng/mL TGF-β2 (R&D Systems, Minneapolis, MN, USA) for 48 h. To minimize variations caused by sample selection, we used match-paired epithelial explants of each eye from the same donor as treatment and control. Samples from age-matched donors were also adopted as biological replicates for statistical analysis. In total, 6 samples from three age-matched donors (annotated 48y_Con, 48y_T2, 52y_Con, 52y_T2, 56y_Con, and 56y_T2) were used for RNA-Sequencing. The human lens epithelial cell line SRA01/04 was kindly offered by Professor Fu Shang of Tufts University (Boston, MA, USA) and cultured in Dulbecco’s Modified Eagle’s medium (DMEM, Thermo Fisher Scientific, Waltham, MA, USA) with 10% FBS, with or without TGF-β2 (5 ng/mL).

### 2.2. In Vitro Transfection with siRNA and Infection with Lentiviruses

The human epithelial explants and SRA01/04 cells were transfected with silencing RNAs (siRNAs) targeting *H19* and nontargeting siRNAs as a negative control (Ribobio, CHN): si-h-H19-001: CCCACAACATGAAAGAAAT; si-h-H19-002: GCCTTTGAATCCGGACACA; si-h-H19-003: GCACTACCTGACTCAGGAA. Following the manufacturer’s instructions, siRNAs were mixed with the OPTI-MEM buffer and transfection reagents (Lipofectamine 3000; Invitrogen, Waltham, MA, USA) to a final concentration of 100 nM. Cells at 70% confluency were used for transfection. At least 24 h of transfection was required for maximal transfection efficiency. For the gain-of-function studies, a specific lentivirus (LV, HANBIO., Shanghai, CHN) carrying a full-length sequence of *H19* (o*H19*) was added into the culture medium at a multiplicity of infection (MOI) of 30 for at least 24 h. Its specific ZsGreen expression detected under fluorescent microscopy guaranteed the transfection efficiency.

### 2.3. RNA Isolation and Analysis

Total RNA from the human lens epithelial explants was collected and purified using the Arcturus PicoPure RNA isolation kit (KIT0204, Applied Biosystems, Waltham, MA, USA) following the manufacturer’s manual. Bio-Rad digital droplet Polymerase Chain Reaction (ddPCR) was then used for RNA analysis. Based on water-oil emulsion droplet technology, each sample was fractionated at the endpoint of amplification into nearly 20,000 nanoliter-sized droplets enclosing RNA inside or not, annotating ‘true’ or ‘false’ events, and PCR amplification took place in each droplet. Absolute RNA concentrations were then calculated (copies/μL).

RNA extraction from SRA01/04 cells was performed with an RNeasy Mini kit (74106, Qiagen, Hilden, Germany), complying with the manufacturer’s protocol. For protein-coding genes and *H19*, PrimeScript IV 1st strand cDNA Synthesis Mix (6215A, Takara, Canton, MA, USA) was used to synthesize first-strand cDNA from total RNA. Quantitative Reverse Transcription-PCR (qRT-PCR) was conducted using QuantiNova SYBR Green PCR (208054, Qiagen, Hilden, Germany) on a QuantStudio 7 Flex system in default thermal cycling settings. Regarding validation of miR-675-3p expression, the microRNA Reverse Transcription Kit was used for adding Poly (A) tail and reverse transcription (EZB-miRT4). EZ-probe qPCR Master Mix for miRNA was used for quantification (EZB-miProbe-R2). The RNA expression levels were calculated using the 2ˆ(-delta delta CT) method. The sequences of primers used in this study are listed in Table 1.

### 2.4. Western Blotting and Capillary Electrophoresis Immunoassay (Wes)

Lens epithelial explants and SRA01/04 cells were lysed with radioimmunoprecipitation assay (RIPA) buffer supplemented with PhosStop Phosphatase Inhibitor Cocktail (Roche Diagnostics, Mannheim, Germany. A BCA assay measured protein concentrations in each sample to normalize the protein loading. To assess the protein levels in lens epithelial explants more accurately, capillary electrophoresis immunoassay, or Simple Western (Wes), was performed on an automated capillary electrophoresis-based platform (ProteinSimple), according to the manufacturer’s protocol (Protein Simple, San Jose, CA, USA). This cutting-edge method is considered more sensitive and accurate than traditional Western blotting, generating highly reproducible results. The Wes 66–440 kDa separation module (Cat. No.: SM-W006) was used. Briefly, protein extracts from each sample were diluted with sample buffer to a final concentration of 2 ug/uL and added to a master mix containing dithiothreitol (DTT) and fluorescent molecular weight markers, then heated at 95 °C for 5 min. After plate loading with the preheated samples, fully automated electrophoresis running, antibody incubation, and immuno-signal detection took place in the capillary system. FN, N-Cad, and α-Actinin (loading control) were detected by primary antibodies ab2413 (1:300, Abcam, Cambridge, UK), ab18203 (1:200, Abcam), and Rabbit anti-α-actinin (1:200, CST 6487S), respectively. Goat Anti-Rabbit horseradish peroxidase (HRP) conjugated secondary antibodies (ready-to-use, DM-001) were used according to the manufacturer’s instructions (ProteinSimple). Compass software (ProteinSimple) generated the digital image visualizing virtual gels and data analysis. Relative protein quantification was generated using α-actinin as an internal control, normalized to the control group.

Traditional Western blotting was performed for protein analysis in SRA01/04 cells. An equal amount of proteins were mixed with 2× SDS-PAGE loading buffer, separated with SurePAGE gels in gradient (4–12%) concentrations (GenScript Biotech Corporation, Nanjing, China), and transferred onto PVDF membranes with pore size 0.45 μm (Merck Millipore, Burlington, MA, USA). The membranes were then blocked in 5% bovine serum albumin (BSA) for 1 h at room temperature (RT) and incubated with primary antibodies (1:1000) overnight at 4 °C. After incubation with secondary HRP-conjugated antibodies (CST, Danvers, MA, USA) at a concentration of 1:2000 for 2 h at RT, protein levels could be visualized with chemiluminescence detection systems (Merck Millipore, IRL). The intensity of each band was determined using Image J software (NIH, Bethesda, MD, USA) and further normalized to GAPDH protein levels. The primary antibodies used in this study are listed in Table 2.

### 2.5. RNAscope/LncRNA In Situ Hybridization

RNAscope kit (Advanced Cell Diagnostics) was used for the in situ detection of *m* transcripts with specifically designed probes (RNAscope probe_Ms-*H19* and RNAscope probe_Hm-*H19*, catalog no.: 423751 and 400771, San Jose, CA, USA). Freshly cut sections from formalin-fixed, paraffin-embedded (FFPE) human and mouse lens tissues, whole-mount of lens epithelial explants, and slides for adherent SRA01/04 cells were pretreated differently following the manufacturer’s instructions. Housekeeping gene *PPIB* was used as a positive control to ensure RNA integrity of the slides (RNAscope probe_Hm-PPIB and RNAscope probe_Ms-Ppib, catalog no.: 18302A and 18346A, San Jose, CA, USA). In contrast, dihydrodipicolinate reductase in Bacillus subtilis (dap B) was adopted as a negative control to rule out the possibilities of false-positive signals. A horseradish peroxidase-based system (RNAscope 2.0 HD Detection Kit-Brown, catalog 310035, San Jose, CA, USA) was adopted for signal amplification, and hematoxylin was used for counterstaining (BA-4097, BASO diagnostics, Zhuhai, CHN). Positive staining was visualized as brown punctate dots.

### 2.6. Immunofluorescence Staining on the Whole Mounts and SRA01/04 Cells, Hematoxylin and Eosin (H&E) Staining on Sections from the Whole Lenses

After treatment, the human lens epithelial explants and SRA01/04 cells were collected. The whole-mount explants were first flattened on a glass slide with tweezers and left half-desiccated. They were fixed with 4% paraformaldehyde for 30 min at RT and washed 3 times with clean PBS. The samples were then saturated with a blocking solution of PBS containing 0.3% Triton X-100 and 3% BSA for 30 min at RT, followed by overnight incubation with primary antibodies at 4 °C. The next day, after three washes in PBS with 0.1% Tween-20 (PBST), lens epithelial cells and explants were incubated with secondary antibodies for 1 h at RT and counterstained with DAPI. The primary antibodies applied are listed in Table 2.

Human and mouse whole lenses were fixed by 10% neutral buffered formalin, dehydrated by gradient ethanol, and embedded into paraffinized blocks. Freshly cut sections were first deparaffinized and rehydrated. Then the nuclei were stained with hematoxylin solution, rinsed with running tap water, and then counterstained with Eosin solution to delineate the cellular cytoplasm (BA-4097 and BA-4098, BASO diagnostics, Zhuhai, CHN).

### 2.7. Cell Scratch Assay

SRA01/04 cells were seeded into 6-well plates and transfected with *H19* silencing reagents or *H19* overexpression lentiviruses. After their concentration reached a 100% confluence, cells were starved for at least 12 h before scratch. With 200 μL pipette tips, the monolayered cells were scratched vertically, leaving a cell-free area (wound) in each well, and detached cells were removed with a PBS wash. Cells were cultured in a serum-free medium to exclude the effects of proliferation, with or without TGF-β2. The migration of cells into the gap was observed and imaged at the start time of 0 h, 12 h, 16 h, and 24 h using a transmitted light microscope. Lines at regular intervals were drawn on the bottom of the plate before starting the experiment to ensure that the exact locations were chosen for observation at distinct time points. The healed areas were measured and quantified using Image J software.

### 2.8. Transwell Migration Assay

After transfection with *H19*-targeting silencing RNAs, or infection with lentiviruses overexpressing *H19* for at least 24 h, SRA01/04 cells were then resuspended at a density of 1 × 10^5^/mL. Cells in 100 μL serum-free medium with or without TGF-β2 were seeded onto the individual upper chamber of 24-well Transwell plates (1 × 10^4^/well) and allowed to migrate vertically through the 8.0 μm pores (Corning, NY, USA). Every lower chamber contained 600 μL DMEM with 10% FBS as the chemoattractant. After 36 h of incubation, the remaining cells on the top of the inserts (non-migrated) were removed with a cotton swab. Cells that migrated to the lower bottom were then fixed with 4% paraformaldehyde (PFA) and detected with Crystal Violet (Beyotime, Shanghai, China). Migrated cells of five randomly selected fields in each well were counted.

### 2.9. Semi-In Vivo Whole Lens Culture

All procedures involving animals were performed following the Association for Research in Vision and Ophthalmology (ARVO) Statement for the Use of Animals in Ophthalmic and Vision Research, and were approved by the Animal Use and Care Committee of Zhongshan Ophthalmic Center (Permit number: 2020-039; Approval date: 3 September 2020). Whole lenses of 21-day-old mice maintained in the C57BL/6J background were collected and cultured in serum-free M199 medium supplemented with 0.1% BSA, 0.1 μg/mL L-glutamine, 100 IU/mL penicillin, and 100 IU/mL streptomycin [33,34]. Whole lenses from human donors were also freshly collected and cultured with the same media. After isolation, lenses were kept in the media for at least 24 h before any treatment. Those with cloudiness caused during surgical collection were excluded. After transfection or infection for 24 h, TGF-β2 (R&D Systems) was added to the medium (10 ng/mL). The medium was changed every other day. Mouse whole lenses were cultured for up to 6 days and photographed under a stereoscope before being harvested. Human whole lens explants were cultured for up to 10 days. RNA levels of *H19* in lens epitheliums were further detected to ensure the efficiency of silencing or overexpressing agents.

### 2.10. RNA Sequencing (RNA-Seq)

Human lens explants were isolated and cultured with or without 10 ng/mL TGF-β2 for 48 h. SRA01/04 cells were first infected with lentiviruses for at least 24 h and then treated with 5 ng/mL TGF-β2 for another 24 h. Total RNA was isolated, and RNA integrity was assessed using a commercial kit and system (RNA Nano 6000 Assay Kit of the Bioanalyzer 2100, Santa Clara, CA, USA). Sequencing libraries were generated using a prep kit for Illumina (NEBNext Ultra RNA Library Prep Kit; NEB). High throughput bulk RNA-sequencing was conducted with a high sequence platform (NovaSeq 6000; Illumina, San Diego, CA, USA) to produce the raw read. Raw sequencing data reported in this paper have been deposited (PRJCA010973) in the Genome Sequence Archive (Genomics, Proteomics & Bioinformatics 2021) in the National Genomics Data Center (Nucleic Acids Res 2022), China National Center for Bioinformation/Beijing Institute of Genomics, Chinese Academy of Sciences, under the Accession codes HRA002775 for human lens epithelial explants and HRA002785 for SRA01/04 cells, and are accessible upon request at https://ngdc.cncb.ac.cn/gsa-human (accessed on 10 August 2022). The Perl script filtered the original data (Raw Reads) and generated Clean Reads to guarantee the data quality used for further analysis. As for the alignment, the reference genomes and the annotation file were first downloaded from the ENSEMBL database. Bowtie2 v2.2.3 was used for building the genome index, and TopHat v2.0.12 was adopted for gene mapping. Reads Count for each gene in each sample was counted by HTSeq v0.6.0. in RPKM (Reads Per Kilobase Million Mapped Reads), which was then calculated to estimate the expression level of genes to eliminate the effects of sequencing depth and gene length.

DESeq2 was used for differential gene expression analysis between two samples with biological replicates, using a model based on the negative binomial distribution. The *p*-value was adjusted by Benjamini and Hochberg’s approach for controlling the false discovery rate. Genes with adjusted *p* ≤ 0.05 and |log2_FC| ≥ 1 were identified as differentially expressed genes (DEGs). Metascape with default parameters [35] was applied for GO enrichment analysis. An Official GSEA software (version 4.0.3, a joint project of UC San Diego and Broad Institute, San Diego, CA and Cambridge, MA) was adopted with weighted enrichment statistics and a *t*-test ranking metric for Gene Set Enrichment Analysis (GSEA) [36]. Gene Set Variation Analysis (GSVA) R package was used to identify the differential signal pathway enrichment [37].

### 2.11. In Vivo Photography with Slit Lamp and Whole Lens Imaging with Stereoscope

*H19* knockout mice were kindly offered by Professor Zhao Meng Lab (Sun Yat-Sen University) [38,39]. The mice were maintained in the C57BL/6J background. The BX 900 slit lamp (Haag-Streit, Mason, OH, USA) was used for a thorough ocular examination of the wild-type and *H19* knockout mice. Immediately after safe anesthesia with intraperitoneal injections of chloral hydrate (5 mL/Kg), their pupils were dilated with 0.5% tropicamide/0.5% phenylephrine mixed eye drop (Santen, Osaka, Japan). A diffusing filter with a wide-open slit-lamp beam at 45 degrees (diffuse illumination) was used to overview the eye appearances. For more detailed information, a narrow focal slit beam (<0.5 mm) and a moderate slit (2–4 mm) was projected at a 60° angle to the lens which directly illuminated the lens pathology. An indirect form of retro-illumination mode was also used: A decentred wide slit beam, adjusted to be square-like, was projected near the pupil margin to reflect light from the fundus, showing the defects in the reflective media along the visual axis. To prevent acute lens opacity induced by anesthesia, we adopted carbomer eye drops, and each eye was examined within 15 min after confirmed anesthesia [40].

Mouse and human lenses were dissected from freshly enucleated eyeballs and washed in warm PBS with antibiotics three times. While still immersed in PBS, the whole lenses were imaged under a Leika stereoscope (S8AP0, Teaneck, NJ, USA) with a high-fidelity digital camera.

### 2.12. Co-Transfection and Luciferase Assay

The putative miRNA binding sites of *SMAD4* were predicted by DIANA-microT v5.0 [41]. Co-transfections of pMIR-REPORT Luciferase-h-*SMAD4* 3′UTR-WT (*SMAD4*-WT, wild type), pMIR-REPORT Luciferase-h-*SMAD4* 3′UTR-MUT (*SMAD4*-MUT, mutant type), combined with miR-675-3p mimics, and miR-675-3p negative control (NC), were completed by Lipofectamine 3000 (Life Technologies, Carlsbad, CA, USA) in SRA01/04 cells. At 48 h of co-transfection, luciferase activities of SRA01/04 cells were measured with a Dual-Luciferase Reporter Gene Assay Kit (Beyotime, Shanghai, China), following the manufacturer’s instructions. Data were further normalized by dividing firefly luciferase activity by renilla luciferase, an internal control to eliminate influences of transfection efficiency.

### 2.13. Statistical Analysis

All experiments were performed at least three times independently. Data were presented as the mean ± standard deviation (SD). The significance of differences between the two groups was determined by the student’s *t*-test, using GraphPad Prism software version 9.0. A one-way analysis of variance (ANOVA) was performed for differences among more than two groups. All statistical tests were two-tailed, and *p*-values less than 0.05 were considered statistically significant (* *p* < 0.05, ** *p* < 0.01, *** *p* < 0.001, **** *p* < 0.0001).

## 3. Results

### 3.1. LncRNA H19 Was Highly Expressed in the Normal Lens Epithelium While Downregulated by Exposure to TGF-β2

Staining LECs in situ on the whole-mount of human lens epithelial explants (LEEs) could better recapitulate lens epithelial features in vivo. In this model, treatment with TGF-β2 resulted in increased extracellular, intracellular, and membranous mesenchymal markers, including FN, α-SMA, and Neural (N)-cadherin (N-Cad), compared to the control group (LEE-Con). At the same time, we observed a loss of tight junction proteins zonula occludens-1 (ZO-1) with cytoplasmic or nuclear translocation of β-catenin, which is typically anchored by Epithelial (E)-cadherin (E-Cad) on cell membranes (Figure 1a). To identify differentially expressed lncRNAs that might play a crucial role in dictating lens epithelial cell fate, we collected the LEEs with or without exposure to TGF-β2 and conducted a high-throughput bulk RNA sequencing. Principal co-ordinates analysis (PCoA) demonstrated dramatic transcriptomic changes in human LEEs upon exposure to TGF-β2 (Appendix A). Over 5000 genes were differentially expressed with statistical significance, including 2828 upregulated and 2147 downregulated ones (Appendix A). Genes decreased by TGF-β2 in LEEs are enriched in GO terms vital to lens development and LEC integrity, such as molecular metabolism, membrane spectrin or channel activities, and oxidation/reduction reactions (Appendix A). Therefore, the competence of LECs was negatively affected by TGF-β2. By contrast, TGF-β2 treatment activated both classical Smad-mediated and non-Smad-dependent downstream signaling pathways. Increased cell proliferation and migration were also implied (Appendix A).

Furthermore, RNA-Seq also identified the top 10 lncRNAs with the most remarkable fold changes across sample replicates (Appendix A). We uncovered that *H19*, the first-discovered lncRNA, was the most highly expressed in human LEEs (Appendix A). Quantitative real-time PCR (qRT-PCR) analysis of lens epithelial explants (age-matched, n = 6) further confirmed that *H19* levels decreased significantly with TGF-β2 treatment (Figure 1b). Using RNAscope (RNA in-situ hybridization) with *H19*-targeting probes, we could delineate the subcellular distribution of *H19* and visualize its relative expression levels. It was self-evident that *H19* expression in the cytoplasm of LECs was significantly reduced under the influence of TGF-β2 compared to the control group (Figure 1c). Quantification data were displayed as absolute values of *H19*-positive areas and relative ratio (Figure 1d,e).

There was a consistently high accumulation of *H19* in the cytoplasm of LECs underneath the anterior capsules from human to adult mouse ocular lenses. Meanwhile, we also demonstrated a conspicuous decrease in *H19* foci and loss of epithelial phenotypes upon exposure to TGF-β2 in semi-in vivo cultures of both human and adult mouse lenses (Figure 1f).

These results indicated that *H19* was indispensable in maintaining the finely-tuned lens epithelial properties. At the same time, the downregulation of lncRNA *H19* might aid in the EMT processes and accelerate the development of lens opacities.

### 3.2. TGF-β2-Reduced H19 Expression in Lens Epithelial Cells in a Time-Dependent Manner

To better understand the role of *H19* during EMT in the ocular lens, the well-established human lens epithelial cell line SRA01/04 was also adopted in this study. Immunofluorescent staining of SRA01/04 cells showed TGF-β2 also led to a more significant amount of mesenchymal markers, including FN, α-SMA, and N-Cad, and drastic morphological changes. Loss of the sub-membranous scaffold protein ZO-1 was observed with an apparent localization shift of β-catenin from cell membranes to the cytoplasm or nucleus (Appendix A). Activation of vital EMT-related transcription factors, such as Snail and Slug, which belonged to the Snail superfamily of zinc-finger transcription factors, were also observed (Appendix A). With RNA probes targeting *H19* hybridized in situ, it was clear that *H19* expression in the cytoplasm of SRA01/04 cells was significantly reduced with TGF-β2 (5 ng/mL, 48 h) in comparison to the control group (con, 0 h) (Appendix A). We further quantified *H19*-positive foci using Image J (Appendix A). Next, we employed EMT markers to validate these alterations at ribonucleic acid (RNA) and protein levels. Treatment of SRA01/04 cells with TGF-β2 (5 ng/mL) increased messenger RNA (mRNA) levels of mesenchymal markers, ranging from FN, and α-SMA, to collagen type 1 (COL-1), along with a reduction in the epithelial junction protein ZO-1, in a time-dependent manner (Appendix A). A duration-dependent decrease in *H19* RNA expression was also observed in SRA01/04 cells in response to TGF-β2 (Appendix A). Protein analysis confirmed that TGF-β2 treatment of SRA01/04 cells gradually increased mesenchymal markers, including FN, α-SMA, N-Cad, and essential transcription factors, Snail and Slug (Appendix A).

Since we had already managed to establish TGF-β2-induced EMT in both SRA01/04 cells and lens epithelial explants, during which a significant reduction in *H19* was observed, the next step was to overexpress or knockdown *H19* forcibly to establish a cause-and-effect relationship.

### 3.3. Knockdown of LncRNA H19 Aggravated TGF-β2-Induced EMT in Both SRA01/04 Cells and Human Lens Epithelial Explants

We first examined the effects of *H19* knockdown in the lens epithelial explants. Due to the enormous variation between each human sample, it is unreliable to pool all capsule samples from different donors and analyze their RNA or protein levels together. However, the amount of RNA or protein collected from every capsule is relatively insufficient for precise and reproducible analysis. Hence, new techniques with higher sensitivity and specificity are needed. The Bio-Rad QX200 droplet digital PCR (ddPCR) system and an automated Simple Western platform (Wes) were thus introduced for RNA and protein analysis for human LEEs. With ddPCR, we measured the absolute RNA copy numbers of *H19* and FN. It was self-evident that FN mRNA levels increased significantly and inversely as *H19* expression dropped (Appendix A). The Wes platform corroborated this at the protein levels of mesenchymal markers FN and N-Cad (Figure 2a–d). Immunofluorescent staining of human LEEs revealed consistent findings between mesenchymal markers FN, α-SMA, N-Cad, and epithelial markers ZO-1 and β-catenin (Figure 2e). More importantly, all these experiments consistently showed that silencing of *H19* alone led to EMT changes comparable to TGF-β2 treatment, which implied that *H19* was necessary for maintaining lens epithelial phenotypes.

Immortalized human lens epithelial cells were also used for functional studies to depict *H19* properties. LncRNA *H19* was forcedly downregulated in SRA01/04 cells by transfection with si*H19* RNAs, among which si*H19*-002 showed the most outstanding silencing effects (Figure 3a). Knockdown of *H19* in cells exposed to TGF-β2 (si*H19*-002+TGF-β2) further increased mRNA levels of FN than TGF-β2 treatment alone (siNC+TGF-β2) (Figure 3b). Western blots confirmed that the downregulation of *H19* led to more production of mesenchymal markers, FN, α-SMA, N-Cad, and abolished the expression of epithelial marker ZO-1 (Figure 3c,d). Consistently, immunofluorescent staining of mesenchymal markers, including FN, α-SMA, N-Cad, and epithelial markers ZO-1 and β-catenin in SRA01/04 cells revealed that silencing of *H19* intensified TGF-β2-induced alterations (Figure 3e). Notably, similar to the observations in epithelial explant studies, silencing of *H19* could also lead to EMT-like alterations, even without the presence of TGF-β2. Furthermore, silencing the lncRNA *H19* plus TGF-β2 treatment group displayed the most significant migrative potential in SRA01/04 cells. It covered almost all of the scratch-induced wound areas within 24 h after the initial scratch (Figure 3f,g). We also used a Transwell assay to confirm this observation, which detected that cells migrated vertically over a certain period (Figure 3h,i). It also demonstrated that knockdown of *H19* alone resulted in a more remarkable migration, either with or without TGF-β2 (Figure 3f–i).

In summary, forced downregulation of *H19* accelerated TGF-β2-induced EMT in human lens epithelial cells with an abnormally active migration. It is worth noting that the knockdown of *H19* could also lead to greater migration capacities and the loss of normal epithelial phenotypes even without TGF-β2 treatment. These results implied that *H19* was necessary for maintaining proper lens epithelial phenotypes.

### 3.4. Overexpression of LncRNA H19 Partially Reversed TGF-β2-Induced EMT and Restored Lens Epithelial Phenotypes

Since *H19* was downregulated during TGF-β2-induced EMT, our next step was to upregulate *H19* expression during EMT and explore its functional roles. We first used a lentivirus vector carrying a full-length sequence of *H19* (o*H19*) to overexpress *H19* in human lens epithelial explants. In contrast, an empty lentivirus vector was applied as a negative control for overexpression (oNC). Human epithelial explants were first infected with lentiviruses, followed by TGF-β2 induction or medium alone.

Then RNA levels of *H19* and FN were measured and quantified by ddPCR. In contrast to what we observed in *H19* silencing studies, upregulation of *H19* partially reversed the transcription of the mesenchymal marker FN, despite the presence of TGF-β2 (Appendix A). We also confirmed EMT changes at protein levels by the Wes platform: *H19* overexpression prevented TGF-β2-induced EMT (Figure 4a–d). Immunofluorescent staining on LEEs confirmed that *H19* reduced the induction of mesenchymal markers FN, α-SMA, and N-Cad, and partially restored membranous ZO-1 and β-catenin (Figure 4e).

Consistent with explant studies, *H19* overexpression in SRA01/04 cells inhibited TGF-β2-induced production in FN and α-SMA mRNA levels (Figure 5a,b). Western blots confirmed the upregulation of *H19*, the attenuated increase in mesenchymal markers, FN, α-SMA, and N-Cad, and the restoration of epithelial marker ZO-1 (Figure 5c,d). Accordingly, immunofluorescent staining of mesenchymal and epithelial markers in SRA01/04 cells proved that o*H19* reversed TGF-β2-induced EMT (Figure 5e). Furthermore, overexpression of *H19* reduced TGF-β2-induced migration in SRA01/04 cells by the scratch assay (Figure 5f,g). Findings from the Transwell assay could draw a similar conclusion (Figure 5h,i).

Briefly, forced upregulation of *H19* maintained LEC phenotypes and partially reversed or prevented TGF-β2-induced EMT in SRA01/04 cells and human lens epithelial explants.

### 3.5. H19 Was Required to Maintain LEC Phenotypes and Lens Clarity

We then moved on to verifying the role of *H19* in the semi-in vivo whole lens studies, which could reflect on its functions more directly. Lenses from 3-week-old mice were first transfected with *H19* silencing agents for 24 h and then incubated with 10 ng/mL TGF-β2 for at least 24 h. Prominent opacities beneath the lens capsules developed, and the knockdown of *H19* extensively aggravated TGF-β2-induced lens fibrosis. Notably, the silencing of *H19* alone without TGF-β2 treatment also induced moderate opacities in the anterior polar region of the lens (Figure 6a). RNA analysis of the epithelial cells collected from mouse whole lenses confirmed the effective knockdown of *H19* and the highest accumulation of FN mRNA in the si*H19* plus TGF-β2 group (Appendix A). Histological examinations revealed that mono-layered cuboidal epithelial cells in those clouded lenses turned into disorganized clumps. Elongated nuclei were observed with an aberrant invasion into the underlying lens fibers (Figure 6b). The opacities beneath the anterior capsules were consistent with an accumulation of FN, α-SMA, and N-Cad, and abolished epithelial junction ZO-1 protein (Figure 6c). Similarly, semi-in vivo cultures of human whole lens explants also revealed that the silencing of *H19* promoted lens fibrosis, while the overexpression of *H19* rescued epithelial features (Figure 7a–e).

### 3.6. H19 Homozygous Knockout Mice Displayed Anterior Polar Cataracts and Delayed Fusion of Anterior Sutures

*H19* homozygous knockout (-/-) mice displayed lens opacities with incomplete genetic penetrance. On Postnatal day 21 (P21), *H19* (-/-) mice exhibited a delayed fusion of anterior sutures with cortical cataracts at the anterior polar regions (Figure 8a–j), consistent with findings observed in semi-in vivo cultured mouse whole lenses with *H19* forcibly silenced. Therefore, it further corroborated that *H19* was vital to constructing lens epithelium and maintaining lens transparency.

### 3.7. Mechanistic Analyses of the Role of H19 in Regulating TGF-β2-Induced EMT in Lens Epithelial Cells

To further understand the potential molecular mechanism, we adopted bulk RNA-Sequencing to mine transcriptomic changes when *H19* was overexpressed in LECs by exposure to TGF-β2 (Appendix A). Briefly, SRA01/04 cells were infected with *H19*-overexpressing lentiviruses or negative control for at least 24 h and further treated with TGF-β2 (5 ng/mL, 24 h). Heatmaps displayed DEGs among different groups, including those responsible for TGF-β2-induced EMT in the lens. It was clear that TGF-β2-induced transcriptomic changes were inversely affected by the overexpression of *H19*. Most genes that TGF-β2 upregulated were reduced with the overexpression of *H19*. By contrast, those with decreased expression with TGF-β2 treatment were mostly rescued/restored by *H19* upregulation (Appendix A). Gene set enrichment analysis (GSEA) validated that gene signature profiles with TGF-β2 induction in LECs were reversed by *H19* overexpression (Appendix A). Furthermore, these genes were involved in vital processes strongly related to EMT, including collagen formation and binding, crosslinking of the collagen fibrils, integrin complex and integrin-mediated cell adhesion, focal adhesion, ECM proteoglycans, and ECM-receptor interaction (Appendix A). Apart from this, vital downstream pathways mediating EMT, including classical Smad-dependent TGF-β receptor signaling, were affected by *H19* (Appendix A).

Specifically, the binding of TGF-β ligands to the TGF-β type I and II serine/threonine kinase receptors causes phosphorylation of downstream receptor-activated Smads (R-Smads), such as Smad2 and Smad3. Their phosphorylated forms (p-Smad2/3) show an increased affinity for the core Smad4 and are shuttled to the nucleus as a heteromeric complex with Smad4 [18,42]. We examined the status of these critical mediators in LECs. As expected, exposure of LECs to TGF-β2 led to considerable phosphorylation of Smad2/3 protein. Upregulation of *H19* reduced TGF-β2-induced Smad2/3 phosphorylation. In contrast, silencing of *H19* in LECs further enhanced p-Smad2 and p-Smad3 levels (Figure 9a–d).

The nuclear translocation of core Smad4 eventually turns on EMT-inducing transcription factors, such as SNAIL and SLUG, ZEB, and TWIST [18,43,44]. Notably, Smad4 expression was also enhanced by the knocking down of *H19* with exposure to TGF-β2, while upregulation of *H19* during TGF-β2-induced EMT reduced Smad4 levels (Figure 9a–d). Interestingly, the 3′-UTR (untranslated region) of *SMAD4* contained predicted binding sites of *miR-675-3p*, which was embedded in the first exon of *H19*. Dual-luciferase assay confirmed that *miR-675-3p* could only bind wild-type *SMAD4* 3′-UTR, not the mutant one (Appendix A), highlighting the potential of the *H19/miR-675-3p/SMAD4* axis in regulating lens fibrosis.

## 4. Discussion

Given that there are currently limited options for lens fibrosis, developing novel strategies or unveiling new targets is of great clinical importance. The prerequisite is gaining an in-depth insight into the molecular mechanisms governing this pro-fibrotic EMT process. This study deciphers the role of lncRNA *H19* in maintaining LEC features by suppressing EMT. We report that the upregulation of *H19* can prevent lens fibrosis induced by TGF-β2 through a blockade of the downstream Smad-dependent signaling.

Less than 3% of genes encode functional proteins in the human genome, while more than 95% are non-coding RNAs (ncRNAs) [45]. Increasing numbers of ncRNAs have been identified, but their functions in biological development and pathological diseases are largely unknown [46,47]. The oncofetal *H19* lncRNA, with relatively high evolutionary conservancy across species, is most abundantly enriched during embryonic stages and in cancer tissues [28,31]. A growing body of studies was conducted on the orchestrating role of *H19* in various cancers in multiple settings, such as initiation, progression, and metastasis [31]. However, controversies remain concerning whether it serves as an oncogene or a tumor suppressor [48,49,50]. Likewise, no consensus has been reached on its promoting or inhibitory role in the pathogenesis of fibrotic diseases [30]. Intriguingly, Raveh proposed a plausible unifying theory that can explain the seemingly contradictory and versatile functions of *H19*, which rely on the actual context [51]. Likely, the exact effect of *H19* on EMT in cancers and fibrotic diseases is tissue- and time-dependent.

We observed for the first time that *H19* was highly-enriched in LECs but downregulated during TGF-β2-induced EMT. Both loss- and gain-of-function studies of *H19* were performed in human lens epithelial explants and SRA01/04 cells, consistently demonstrating that the silencing of *H19* exacerbated EMT. At the same time, the overexpression of *H19* reversed this phenotypic transformation. Noteworthily, knocking down *H19* itself could lead to EMT-like changes, even without TGF-β2, implying the indispensability of *H19* in maintaining LEC integrity. RNA-Seq data further validated that *H19* overexpression could reverse TGF-β2-induced transcriptomic alterations to restore the epithelial phenotypes, indicating the importance of *H19* in maintaining the distinctive properties of LECs.

LECs underlying the anterior lens capsules are drawing increasing attention for their importance in establishing and maintaining lens transparency. Many studies have been conducted to explore the potential therapeutic options for preventing or reversing lens fibrosis using diverse experimental models. To better validate their roles in lens transparency maintenance, in recent years, apart from in vitro studies using lens epithelial cell lines, semi-in vivo epithelial explants, the whole lens culturing, and in vivo animal models have been more widely used, including injury-induced ASC models, mock lens cataract surgery (Extracapsular Lens Extraction, ECLE) for PCO models, and transgenic/genetic knockout mice [19,52,53,54,55,56,57]. It is generally accepted that during TGF-β2-induced EMT, loss of LEC integrity is accompanied by the abnormal secretion of ECM and aberrant cell migration, resulting in fibrotic plaques and lens opacification. Our semi-in vivo whole lens culture studies corroborated that the downregulation of *H19* led to significant EMT of LECs, contributing to clouding of the lens with or without exposure to TGF-β2. On the contrary, the upregulation of *H19* maintained lens epithelial properties and restored lens clarity. Similar lens opacities were observed in *H19* homozygous knockout mice, indicating the indispensable role of *H19* in establishing lens transparency through the maintenance of LEC phenotypes. Apart from fibrotic opacities formed in the lens, EMT of LECs also led to abnormal differentiation, resulting in defective lens fibers with the disordered arrangement. Upregulation of *H19* further avoids the abnormality of lens fibers by restoring LECs, which also contributes to lens transparency. Therefore, *H19* is vital for maintaining LEC phenotypes and sustaining lens clarity.

As a long non-coding RNA, *H19* can interact with DNA/chromatin, RNA, and proteins, regulating EMT at transcriptional, post-transcriptional, and post-translational levels [58,59,60]. Notably, competing endogenous RNA networks (ceRNETs) are gaining increasing attention: *H19* may also execute a decoy for sequestrating miRNAs [60,61,62]. Alternatively, *H19* also serves as a reservoir for producing miRNAs [63]. The first exon of the *H19* gene itself can be transcribed into miRNA-675 as a by-product which targets functional proteins involved in various biological and pathological processes, such as EMT [63,64,65,66,67]. Apart from the predicted binding from the online database, we have confirmed the binding of *miR-675-3p* to 3′-UTR of *SMAD4* directly in LECs with luciferase reporter assays (Appendix A). Smad4 is the core member belonging to the Smads family. It transduces signals from TGF-β2 signaling and activates crucial downstream effector genes via nuclear translocation [8]. Admittedly, when *H19* was upregulated by transfection in LECs, the changes in miRNA-675 levels were much more minor and negligible compared to *H19* (Appendix A). Therefore, alternative explanations are warranted for the regulatory role of *H19* in lens fibrosis.

Cheng reported an increase in *H19* expression in human LECs upon exposure to UV-B, contributing to the oxidative damage of LECs [68]. This study observed the considerable downregulation of *H19* in human and mouse LECs with TGF-β2 administration. Previous studies implied the possible involvement of both canonical Smad-dependent and Smad-independent pathways, such as mitogen-activated protein kinase (MAPK)/extracellular signal-regulated kinase (ERK)1/2, phosphatidylinositol-3-kinase (PI3K)/Akt, and the mammalian target of rapamycin (mTOR) signaling during lens fibrosis [18,42,61]. RNA-Seq of LECs suggested their activation by TGF-β2, whereas *H19* overexpression blocked such changes (Appendix A). Immunoblotting confirmed that upregulation of *H19* reduced phosphorylation of R-Smads (Smad2/3) and inhibited core Smad4 expression directly. Our RNA-seq data also revealed the involvement of essential processes, such as collagen formation and crosslinking, integrin-mediated cell adhesion, focal adhesion, and ECM-receptor interaction. Therefore, future work could be directed at the crosstalks of these critical signaling pathways or biological processes.

In particular, *H19* is unique in its genetic linkage with its neighboring gene, insulin-like growth factor 2 (IGF2), where both genes are expressed reciprocally or parental-specifically [69]. This genetic imprinting at the *H19*/IGF2 locus was reported to be strictly controlled by an upstream differentially methylated domain (DMD), also known as the imprinting control region (ICR) [38,70,71]. However, our current study can temporarily exclude the effects of genetic imprinting, as RNA analysis by qPCR revealed a similar increase in IGF2 mRNA levels when *H19* was overexpressed (Appendix A). We have already obtained genetic animals with *H19*-DMD/ICR deletions and found congenital lens developmental defects in heterozygous knockout mice (Appendix A). Functional studies of these genetic mice are required to clarify the specific role of *H19*/IGF2 balance in lens development.

Admittedly, apart from *H19*, alternative epigenetic regulatory ncRNAs may also be involved in lens fibrosis [32,72,73,74]. Nevertheless, our current study presented the first evidence that *H19* serves a protective role during TGF-β2-induced EMT in the lens. Noteworthily, we demonstrated the close relationship between the integrity of LECs and the maintenance of lens transparency. In conclusion, although further investigations are needed, our findings open up possibilities for *H19* as a novel and potential therapeutic target for treating lens fibrosis.

## 5. Conclusions

This study showed for the first time that lncRNA *H19* could prevent TGF-β2-induced lens fibrosis by suppressing the EMT-mediating canonical Smad-dependent pathways. Overexpression of *H19* could partially restore lens transparency by reversing EMT and maintaining lens epithelial features. Thus, *H19* could be a potential therapeutic option for fibrotic cataracts.

## Figures and Tables

**Figure 1 cells-11-02559-f001:**
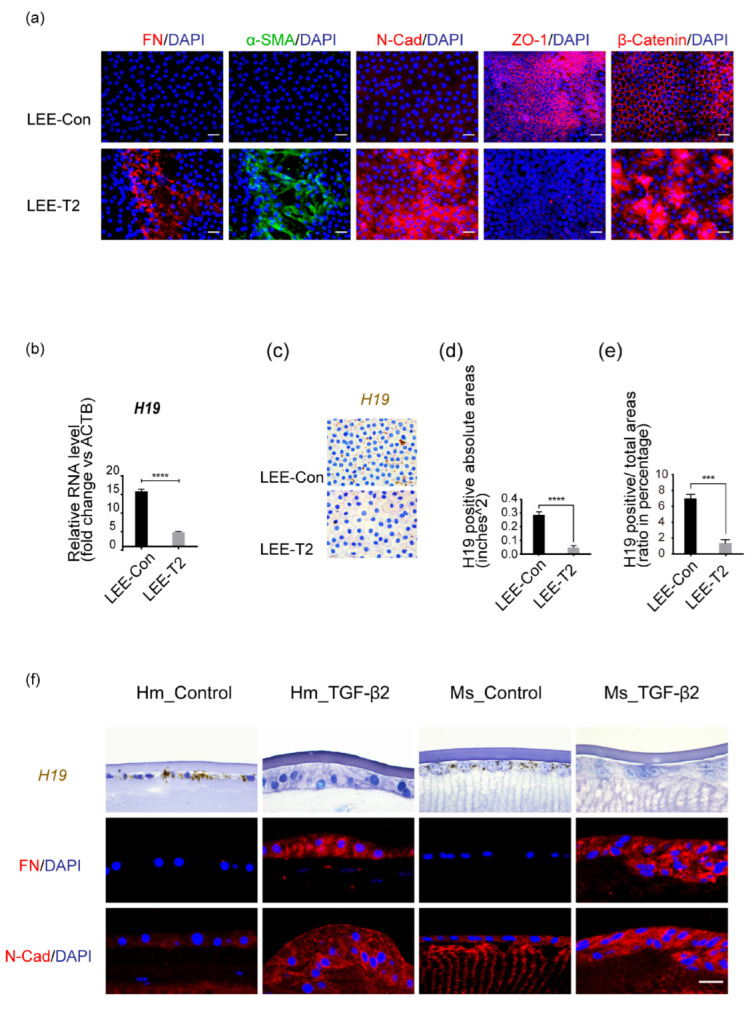
LncRNA *H19* was highly expressed in normal lens epithelium while downregulated by exposure to TGF-β2. (**a**) Immunofluorescent staining of mesenchymal markers (FN, α-SMA, and N-Cad), and epithelial markers (ZO-1, β-catenin), on human lens epithelial explants (LEEs), when exposed to TGF-β2 (10 ng/mL, 48 h; LEE-T2), in comparison to control group (LEE-Con). Scale bars, 20 μm. (**b**) RNA analysis of LEEs-Con and LEEs-T2 (age-matched, n = 6) was performed using β-actin as the internal control. (**c**) *H19* probes were hybridized in situ. Scale bars, 20 μm. (**d**,**e**) Image J was used to quantify the absolute value of *H19* positive areas (**d**) and the ratio of *H19* positive to total area within each cell (**e**). At least three experiments were repeated, and data were shown as mean ± SD. *** *p* < 0.001, **** *p* < 0.0001 vs. LEE-Con. (**f**) RNAscope targeting *H19* on the freshly cut sections from human and mouse whole lenses was performed. The lower two panels showed immunostaining of mesenchymal markers FN and N-Cad. Scale bars, 20 μm.

**Figure 2 cells-11-02559-f002:**
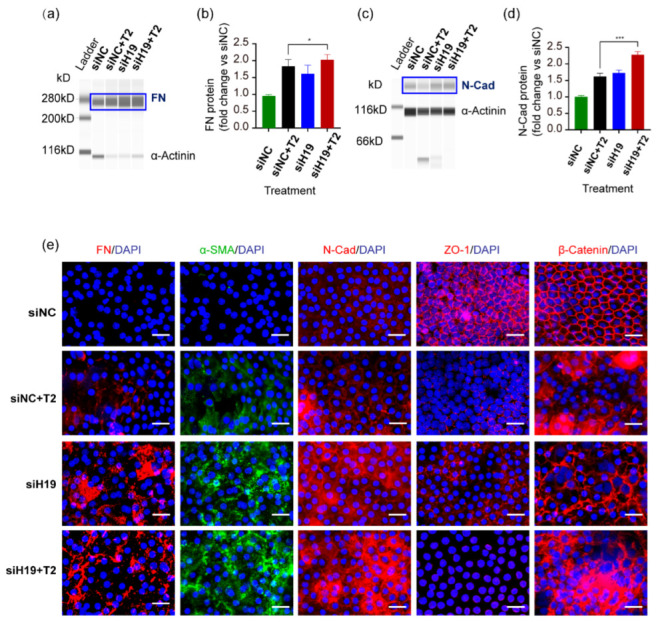
Knockdown of *H19* in human lens epithelial cells in situ accelerated TGF-β2-induced EMT. (**a**–**d**) After transfection with si*H19*-002 for 24 h and further exposure to TGF-β2 for 24 h, the Wes platform detected protein levels of mesenchymal markers FN and N-Cad in the human LEEs, which were further quantified using Image J. * *p* < 0.05, *** *p* < 0.001. (**e**) Fluorescent staining of FN, α-SMA, N-Cad, ZO-1, and β-catenin on human LEEs. Scale bars, 20 μm.

**Figure 3 cells-11-02559-f003:**
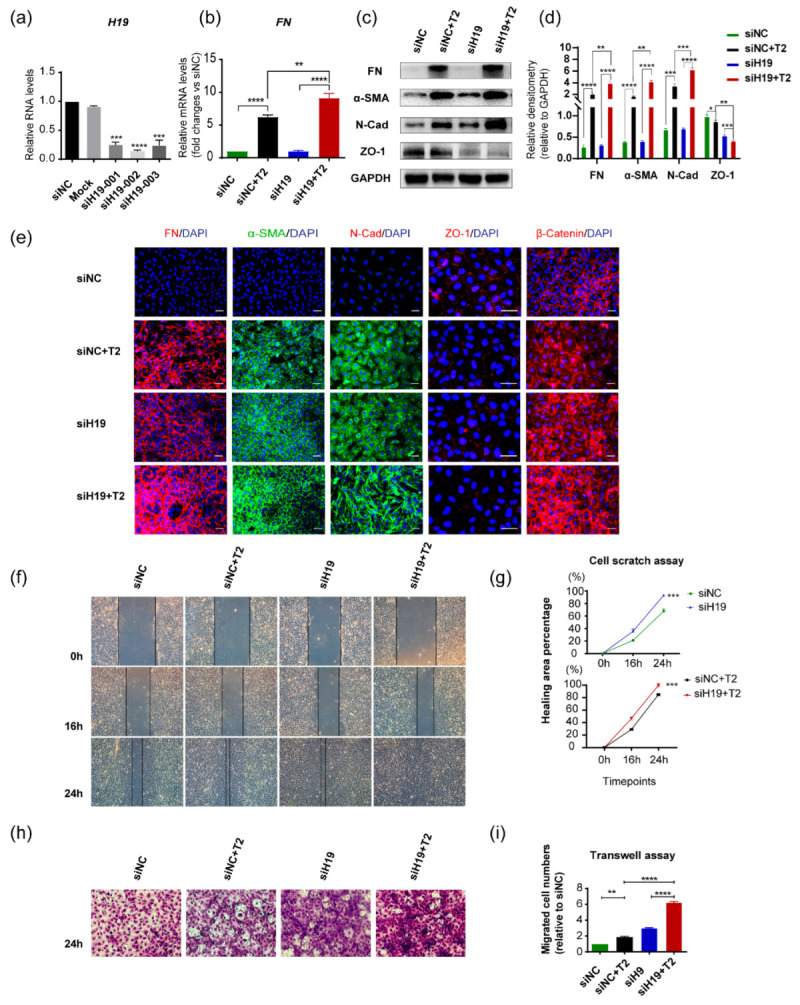
Downregulation of *H19* in SRA01/04 cells aggravated TGF-β2-induced EMT. (**a**) After transient transfection with *H19* siRNAs in SRA01/04 cells for 48 h, *H19* expression was determined by quantitative real-time PCR. (**b**) FN mRNA levels were detected in SRA01/04 cells after transfection with si*H19*-002 for 24 h and further exposure to TGF-β2 for 24 h. (**c**,**d**) Protein levels of mesenchymal markers FN, α-SMA, N-Cad, and epithelial markers ZO-1. Representative western blots and densitometric analyses (mean ± SD) are shown (n= 3 per group). Protein data are expressed as a ratio to GAPDH. (**e**) Staining of FN, α-SMA, N-Cad, ZO-1, and β-catenin in SRA01/04 cells. Scale bars, 50 μm. (**f**–**i**) After transfection with si*H19*-002, the migration capacities of SRA01/04 cells induced by TGF-β2 was detected by cell scratch assay (**f**,**g**) horizontally and Transwell assay (**h**,**i**) vertically. Image J was used for further quantification and calculation. Scale bars, 50 μm. * *p* < 0.05, ** *p* < 0.01, *** *p* < 0.001, **** *p* < 0.0001 vs. control group (siNC).

**Figure 4 cells-11-02559-f004:**
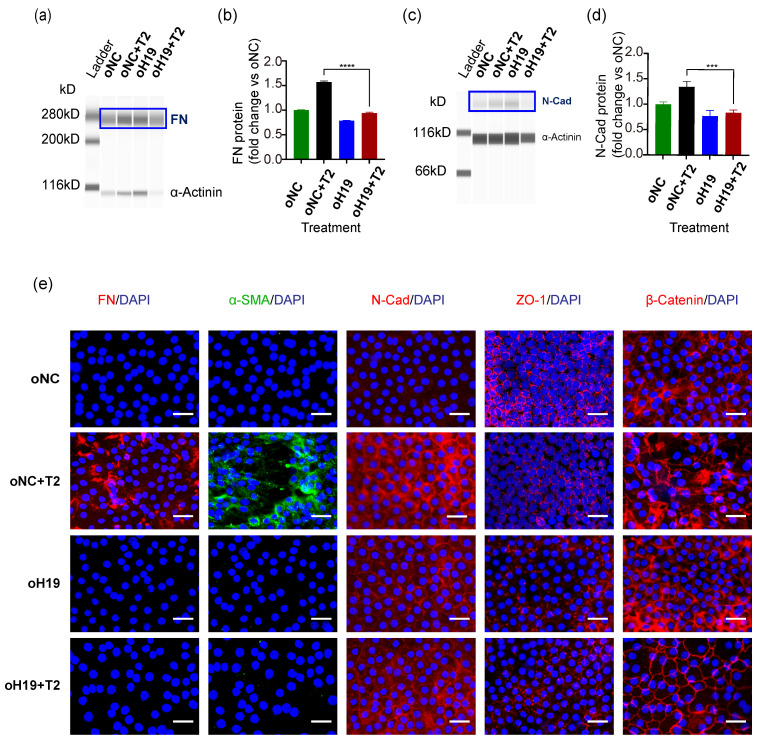
Overexpression of *H19* in human lens epithelial cells in situ prevented TGF-β2-induced EMT. (**a**–**d**) After infection with *H19*-overexpressed lentiviruses for 24 h and further exposure to TGF-β2 for another 24 h, ProteinSimple Wes platform detected mesenchymal markers, FN, and N-Cad protein levels. (**e**) Immunofluorescent staining of FN, α-SMA, N-Cad, ZO-1, and β-catenin on human LEEs. Scale bars, 20 μm. *** *p* < 0.001, **** *p* < 0.0001.

**Figure 5 cells-11-02559-f005:**
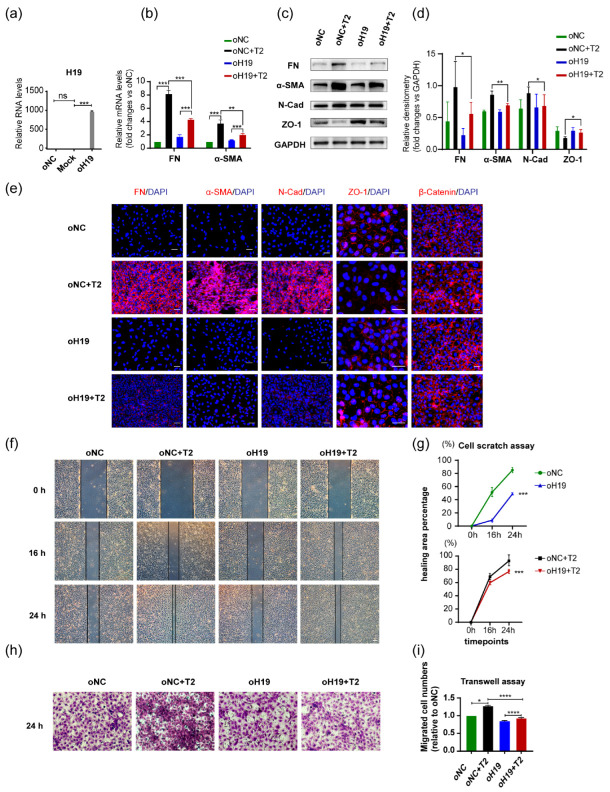
Upregulation of *H19* in SRA01/04 cells partially restored lens epithelial cell properties. (**a**) After SRA01/04 cells were infected with the *H19*-overexpressed lentiviruses for 48 h, *H19* expression was validated by quantitative real-time PCR. (**b**) RNA analyses of FN and αSMA in SRA01/04 cells after infection for 24 h and further exposure to TGF-β2 for 24 h. (**c**,**d**) Protein levels of mesenchymal markers FN, α-SMA, N-Cad, and epithelial cell junction marker ZO-1. Representative western blots are shown here. Protein quantification data are expressed as a ratio to GAPDH. (**e**) Staining of FN, α-SMA, N-Cad, ZO-1, and β-catenin in SRA01/04 cells. (**f**–**i**) After infection with o*H19*, the migration capacity of SRA01/04 cells induced by TGF-β2 was detected by wound-healing assay (**f**,**g**) horizontally and Transwell assay (**h**,**i**) vertically. Image J was used for further quantification. Scale bars, 50 μm. * *p* < 0.05, ** *p* < 0.01, *** *p* < 0.001, **** *p* < 0.0001, ns: not significant. oNC, normal control for overexpression lentiviruses; o*H19*, *H19* overexpression lentiviruses.

**Figure 6 cells-11-02559-f006:**
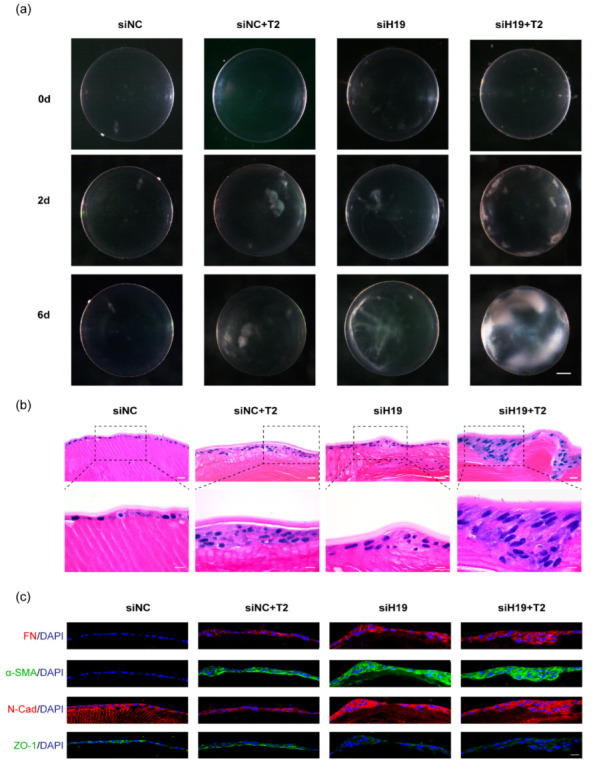
*H19* was required to maintain LECs phenotype and lens clarity in semi-in vivo mouse whole lens cultures. (**a**) Images were taken by stereoscope with a dark background to demonstrate lens clarity. Scale bar, 200 μm. (**b**) Hematoxylin and Eosin (HE) staining of the mouse whole lens explants was collected after being cultured for 6 days. Scale bars, 10 μm. (**c**) Immunofluorescent staining of FN, α-SMA, N-Cad, and ZO-1 in freshly cut sections of the mouse whole lens explants was also collected on day 6 of culturing. Scale bars, 20 μm.

**Figure 7 cells-11-02559-f007:**
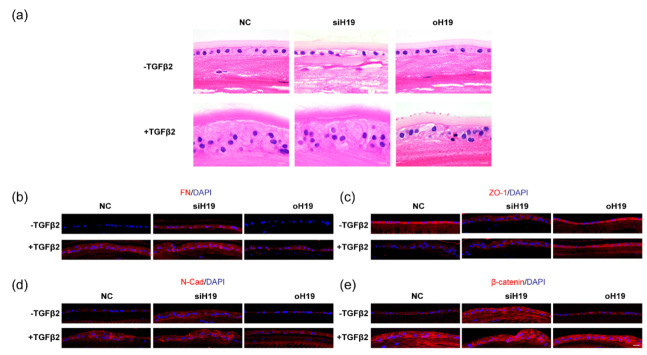
*H19* was required to maintain LECs phenotype and lens clarity in semi-in vivo human whole lens cultures. (**a**) HE staining of freshly cut sections from the human whole lens explants, collected after being cultured with different treatments for 10 days. Scale bars, 10 μm. (**b**–**e**) Immunofluorescent staining of FN (**b**), ZO-1 (**c**), N-Cad (**d**), and β-catenin (**e**). Scale bars, 20 μm.

**Figure 8 cells-11-02559-f008:**
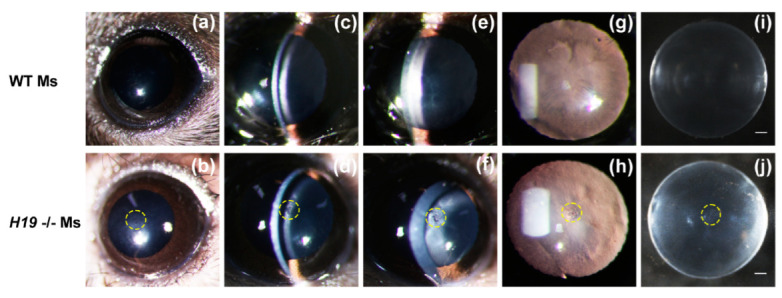
*H19* knockout mice displayed anterior polar cataracts and delayed anterior suture fusion. (**a**–**h**) Photos were taken under slit lamps after the mice were anesthetized and their pupils were dilated. The ocular lenses of 8-week *H19* (-/-) mice and the wild-type littermates were examined in-vivo by Haag-Streit BX 900 slit lamp with a diffuse illumination mode (**a**,**b**), a narrow slit mode (**c**,**d**), a moderate slit mode (**e**,**f**), and a retro-illumination mode (**g**,**h**). (**i**,**j**) Photos of freshly enucleated lenses from 8-week *H19* (-/-) mice and the wild-type littermates. Observations in both eyes were similar, and representative pictures of one eye were shown here. Scale bars, 0.5 mm.

**Figure 9 cells-11-02559-f009:**
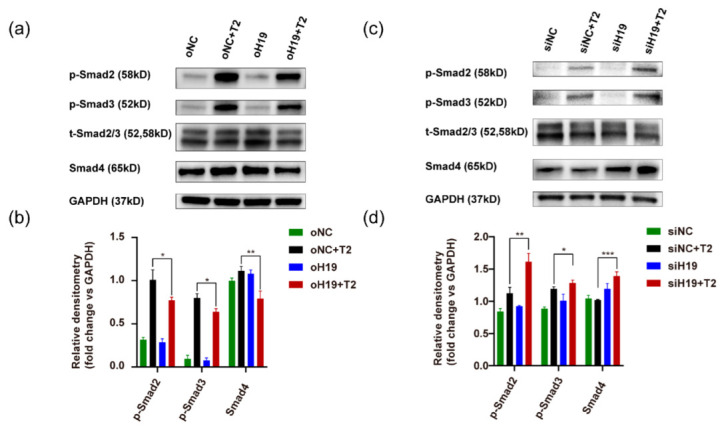
Mechanistic analyses of the role of *H19* in regulating TGF-β2-induced EMT in lens epithelial cells. (**a**–**d**) SRA01/04 cells were first infected with *H19*-overexpressing lentiviruses or negative control for at least 24 h and further treated with TGF-β2 (5 ng/mL, 24 h). Protein levels of p-Smad2, p-Smad3, total Smad2/3, and Smad4 were examined, using GAPDH as an internal control. Representative western blots and densitometric analyses (mean ± SD) are shown (n = 3 per group). * *p* < 0.05, ** *p* < 0.01, *** *p* < 0.001.

**Table 1 cells-11-02559-t001:** Nucleotide sequences of primers used in this study.

Species	Genes	Forward	Reverse
Mouse	*FN1*	TTACCAACCGCAGACTCACC	CCACTGCATTCCCACAGAGT
	*H19*	CTAGGCTGGGGTCAAACAGG	TTACGGTGGGTGGGATGTTG
	*Actb*	TGAGCTGCGTTTTACACCCT	GCCTTCACCGTTCCAGTTTT
Human	*H19*	CGGACACAAAACCCTCTAGC	GCTGTTCCGATGGTGTCTTT
	*FN1*	GAGCTGCACATGTCTTGGGAAC	GGAGCAAATGGCACCGAGATA
	*ACTA2*	AAAAGACAGCTACGTGGGTGA	GCCATGTTCTATCGGGTACTTC
	*COL1A2*	CCTCAAGGTTTCCAAGG	CACCCTGTGGTCCAACAACTC
	*TJP1*	TGGTCTGTTTGCCCACTGTT	TCTGTACATGCTGGCCAAGG
	*IGF2*	CGTCCCCTGATTGCTCTACC	CGGCAGTTTTGCTCACTTCC
	*GAPDH*	GGAGTCCACTGGCGTCTTCA	GTCATGAGTCCTTCCACGATACC
	*ACTB*	CATTCCAAATATGAGATGCGTT	TACACGAAAGCAATGCTATCAC
	*miR-675-3p*	CTGTATGCCCTCACCACTCA	
	*U6*	CAAGGATGACACGCAAATTCG	

**Table 2 cells-11-02559-t002:** Primary antibodies used in this study.

Host Species	Proteins	Company	Cat. No.
Rabbit	Fibronectin (FN)	Abcam (Cambridge, UK)	ab2413 (for WB, WES)
		Abcam (Cambridge, UK)	ab137720 (for IF/ICC)
Mouse	α-smooth muscle actin (α-SMA)	Abcam (Cambridge, UK)	ab7817
Rabbit	N-Cadherin(N-Cad)	Abcam (Cambridge, UK)	ab18203
Rabbit	Zonula Occludens-1(ZO-1)	Thermo Fisher Scientific (Rockford, IL, USA)	61-7300
Rabbit	β-Catenin	Cell Signaling Technology (Danvers, MA, USA)	CST8480
Rabbit	Snail/Slug	Abcam (Cambridge, UK)	ab180714
Rabbit	Phospho-Smad2	Cell Signaling Technology (Danvers, MA, USA)	CST3108
Rabbit	Phospho-Smad3	Cell Signaling Technology	CST9520
Rabbit	Smad4	Abcam (Cambridge, UK)	ab40759
Rabbit	Smad2/3	Cell Signaling Technology (Danvers, MA, USA)	CST8685
Rabbit	GAPDH	Cell Signaling Technology (Danvers, MA, USA)	CST2118
Mouse	β-Actin	Cell Signaling Technology (Danvers, MA, USA)	CST3700
Rabbit	α-Actinin	Cell Signaling Technology (Danvers, MA, USA)	CST6487S

## Data Availability

The datasets generated in this study are not publicly available but can be accessed from the corresponding author upon reasonable request.

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
