# Peer review of "Long Non-Coding RNA H19 Prevents Lens Fibrosis through Maintaining Lens Epithelial Cell Phenotypes"

_cells, 2022, doi:10.3390/cells11162559_

Round 1

Reviewer 1 Report

This is a very well designed study that provides solid new information about potential mechanisms driving LEC EMT following lens injury/cataract surgery.

However, there are several issues that need to be addressed.

The most important one is that the RNAseq data described in the manuscript does not seem to have been submitted to a public respository such as GEO so that the field can assess the data and reuse it for other purposes.  The Geo accessions should be given in the manuscript.  As part of this, the manuscript does not state the number of biological replicates for this study which is essential, particularly (as pointed out by the authors) human primary tissue can be variable.

The human tissue procurement needs clarification as the lenses are stated to be donor globes but the methods state that the tissue was obtained from consented subjects.  How do you obtain "written informed consent" from each subject involved as dead people would presumably be the source of donor lenses?

There are also a bunch of missing methods.  

How were the organ culture experiments with H19 knockdown done?  How did the vectors get accross the lens capsule. No details are given

The culture experiments done for RNAseq give no details about time in culture after initial isolation and how long post treatment and how many biological replicates done.

 There is no info in the methods on the H19 null mice, their strain, references for how made etc, where obtained etc

The manuscript does not cite much of the recent literature on the mechanisms of LEC EMT that have been reported in mice and chick models in the past 3 years or so.  

Author Response

Thank you for providing us valuable advice and bringing up insightful concerns which helped improve our manuscript greatly. Please see the attachment.

Reviewer 2 Report

In this manuscript the authors have conducted a very comprehensive investigation into the role played by the long 16 non-coding RNA (lncRNA) H19 in regulating TGF-β2-induced EMT during lens fibrosis. They show that lncRNA H19 is highly expressed in LECs but downregulated by exposure to TGF-β2. Using both human lens epithelial explants and SRA01/04 cells they show that knockdown of H19 aggravated TGF-β2-induced EMT, while overexpressing H19 partially reversed EMT and restored lens epithelial phenotypes. This work was supplemented by whole lens culture experiments and the development of an H19 knockout mouse. Finally, the authors showed that the action of H19 was mediated Smad-dependent signalling pathways.

This was a very robust study and I have no problems with any of the results or subsequent conclusions. I do however have some editorial comments that focus on the fact that I found the volume of data presented in the manuscript some what overwhelming to the extent that the volume of results presented tended to dilute the important major findings of the study. I suppose my message is less is more.

To address my concerns, I think the authors should consider moving more of their result into supplementary figures and focus on the main results. This would allow them to increase the size of some of their images to more clearly display their data. For example

Figure 1. Move panels b-g to supplementary data and reformat figure to increase the size of panels i-l.

Figure 2 & 4 Delete panels a &b and include results in the text. Reformat figure and increase the size of panel g

Figure 6 Delete panels b & c and include results in the text.

Figure 9 Move panels a – e to supplementary data

Author Response

We appreciate your insightful advice and agree on the principle of "less is more". Based on your suggestions, we made revisions to our figures to better present the results/findings. Please see the attachment for more information.

Reviewer 3 Report

This is a very good study exploring biological roles of H19. The authors used appropriate methodologies and presented the findings clearly.

Authors should check the manuscript for minor typos and grammar errors. Some examples:

Line 95. Please re-write the sentence.

Line 153. western >> Western.

Line 210. 1x105? Please check.

Line 305. hu-man >> human.

Line 387. plat-form >> platform

Figure 6d. The area of magnification should be shown in the upper panel.

Author Response

We sincerely appreciate your time and effort for reviewing and thanks for your helpful suggestions. We made corresponding revisions in the latest manuscript. Please see the attachment for more information.

Round 2

Reviewer 2 Report

I think the results are a lot less cluttered now. You could however still look at increasing the size of some images within you figures to make sure that they are the optimal size to fill the space available i.e. in Figures 2-5 look at making panels a-b and c-d underneath each other rather than in a line across the page

This manuscript is a resubmission of an earlier submission. The following is a list of the peer review reports and author responses from that submission.